# Link Worker social prescribing to improve health and well-being for people with long-term conditions: qualitative study of service user perceptions

Suzanne Moffatt, Mel Steer, Sarah Lawson, Linda Penn, Nicola O'Brien

Institute of Health and Society, Newcastle University, Newcastle upon Tyne, Tyne and Wear, UK

**Correspondence to**
Dr Suzanne Moffatt; suzanne.moffatt@ncl.ac.uk

## ABSTRACT

**Objectives** To describe the experiences of patients with long-term conditions who are referred to and engage with a Link Worker social prescribing programme and identify the impact of the Link Worker programme on health and well-being.

**Design** Qualitative study using semistructured interviews with thematic analysis of the data.

**Intervention** Link Worker social prescribing programme comprising personalised support to identify meaningful health and wellness goals, ongoing support to achieve agreed objectives and linkage into appropriate community services.

**Setting** Inner-city area in West Newcastle upon Tyne, UK (population n=132 000) ranked 40th most socioeconomically deprived in England, served by 17 general practices.

**Participants** Thirty adults with long-term conditions, 14 female, 16 male aged 40–74 years, mean age 62 years, 24 white British, 1 white Irish, 5 from black and minority ethnic communities.

**Results** Most participants experienced multimorbidity combined with mental health problems, low self-confidence and social isolation. All were adversely affected physically, emotionally and socially by their health problems. The intervention engendered feelings of control and self-confidence, reduced social isolation and had a positive impact on health-related behaviours including weight loss, healthier eating and increased physical activity. Management of long-term conditions and mental health in the face of multimorbidity improved and participants reported greater resilience and more effective problem-solving strategies.

**Conclusions** Findings suggest that tackling complex and long-term health problems requires an extensive holistic approach not possible in routine primary care. This model of social prescribing, which takes into account physical and mental health, and social and economic issues, was successful for patients who engaged with the service. Future research on a larger scale is required to assess when and for whom social prescribing is clinically effective and cost-effective.

## Strengths and limitations of this study

► This is the first UK study to provide detailed insight into the impact of a Link Worker social prescribing programme on health and well-being among adults with long-term conditions living in an area of high socioeconomic deprivation.

► Our interview data showed that the Link Worker social prescribing programme engendered feelings of control and self-confidence, reduced social isolation and led to positive physical and behavioural changes such as weight loss, increased physical activity, improved mental health and long-term condition management as well as greater resilience and effective coping strategies to manage relapses.

► Key elements of the Link Worker social prescribing model are that it: addresses the coexistence of multimorbidity, mental health problems and social isolation; is long-term in nature; and, where applicable, tackles related socioeconomic issues.

► A strength of this work was the depth and consistency of participant accounts regarding the impact of the programme. Limitations were that we did not capture the experiences of patients with long-term conditions who refuse a referral or who drop out of the programme early, nor were we able to collect information on the frequency with which participants engaged in activities they were referred to.

► Our study adds to the emerging evidence base on Link Worker social prescribing by demonstrating improvements in health-related behaviours and long-term condition management.

## BACKGROUND

Social prescribing enables healthcare practitioners to refer patients to a range of non-clinical services.[1] Primarily, but not solely, directed at people with long-term conditions, social prescribing harnesses assets within the voluntary and community sectors to improve and encourage self-care and facilitate health-creating communities.[2–4] There is increasing interest in social prescribing as a means of addressing complex health, psychological and social issues presented in primary

care, as well as its potential to reduce health inequalities.[5] A recent review of social prescribing indicates that, despite a small and largely inconclusive evidence base, there is the potential for credible psychosocial benefits to be incurred by patients with mental health problems, and for health and well-being improvements to be seen in people with long-term conditions.[6] While less attention has been paid to the impact of social prescribing on physical health and resource use,[4] improvements in physical activity,[7] reductions in hospital resource use[8] and General Practitioner (GP) attendance[9] have been attributed to social prescribing, although longer-term studies with larger sample sizes are required for more definitive evidence.[4]

As yet there is no agreed definition of social prescribing,[2] although there is broad consensus that it helps patients to access non-clinical sources of support, predominantly in the community sector,[3] and is a means to address the well documented social and economic factors that accompany long-term illness beyond the healthcare setting.[4] In the UK, social prescribing has been taking place on a small scale for a number of years and there are several operating models.[10] These models vary in two ways; the actual activities or services offered and the level of support given to patients following referral. Recognising that patients who are simply given information about a service will not necessarily take it up, most schemes involve a 'facilitator' coupled with personal support,[11] although the level of ongoing support offered varies considerably. Services into which patients are referred vary, and can include activities that involve physical activity such as gyms, walking groups, gardening or dance clubs; weight management and healthy eating activities, such as cooking clubs. Addressing wider economic and social issues can involve referral into services which address welfare, debt, housing and employment issues. Groups, such as those targeted at people with specific long-term conditions, for example diabetes, chronic obstructive pulmonary disease, may also be accessed via social prescription. Our definition concurs with that of the Social Prescribing Network of Ireland and Great Britain, 'enabling healthcare professionals to refer patients to a link worker, to co-design a non-clinical social prescription to improve their health and well-being' and use services provided by the voluntary and community sector. (p19)[2]

Ways to Wellness[12] is one of the first UK organisations to deliver social prescribing on a large and prolonged scale, funded for 7 years through a social impact bond model, with an overall target of 11 000 users over this period. Based in west Newcastle upon Tyne, an area with some of the most socioeconomically deprived wards in England, Ways to Wellness covers 17 general practices. Referral criteria are men and women aged 40–74 years with one or more of the following long-term conditions: diabetes (types 1 and 2), chronic obstructive pulmonary disease, asthma, coronary heart disease, heart failure, epilepsy, osteoporosis, with or without anxiety or depression. Ways to Wellness is delivered by four voluntary sector organisations.

Following extensive consultation with patients and healthcare professionals over an 8-year period,[13] Ways to Wellness provides a 'hub' model of social prescribing in which a Link Worker trained in behaviour change methods offers a holistic and personalised service. Following referral from a primary care practitioner (GP, practice nurse, healthcare assistant), meaningful health and wellness goals are jointly identified and service users are connected, when desired, to community and voluntary groups and resources. The service comprises: (A) individual assessment, motivational interviewing and action planning; (B) completion of an initial 'Well-being Star' assessment and subsequent Well-being Star assessments every 6 months thereafter for the duration of the patient's involvement; (C) help to access community services (eg, welfare rights advice, walking groups, physical activity classes, arts groups, continuing education); (D) promotion of volunteering opportunities, and; (E) promotion of improved self-care and sustained behaviour change related to healthier lifestyle choices. Thus, the programme is highly individualised with patient engagement varying in terms of intensity, duration, personalised goal-setting and onward referral. Patients can remain with the programme for up to 2 years, but with Link Worker discretion beyond 2 years if required; frequency of contact with the Link Worker is mutually agreed, varies between and within patients depending on current need and circumstances, and can be face to face, via telephone, email and/or text message. Data for this study were collected in the first 14 months of the Link Worker social prescribing programme implementation.

This qualitative study aimed to capture the experiences of patients engaged with Ways to Wellness in its first 14 months of operation and to identify the impact of the Link Worker social prescribing programme on health and well-being.

## METHODS
### Setting
This study was set in an inner-city area of high socioeconomic deprivation in the west of Newcastle upon Tyne (population n=132 000), ranked among the 40 most deprived in England according to the Index of Multiple Deprivation.[14] Eighteen per cent of residents have long-term conditions and receive sickness and disability-related benefits, which is higher than the national average.[15]

### Recruitment and sampling
The four Ways to Wellness provider organisations acted as gatekeepers and approached service users on our behalf, explaining the study and issuing a participant information sheet. Those who agreed for their details to be passed on to the researcher were contacted by telephone to ascertain willingness to participate, and, if willing, an interview was arranged. We set out to obtain a maximum variation sample across the four provider organisations based on the following criteria: age, gender, long-term condition,

marital status, employment status, socioeconomic status and level of engagement with Ways to Wellness (intensive to non-intensive). For the purposes of sampling, long-term condition was ascertained by the Link Worker via the referral form, and had therefore been diagnosed by a GP or other healthcare professional. The recruitment period was January to June 2016. Ways to Wellness was operational from April 2015, and in the 14 months since it started, 864 women and 739 men, average age 59 years, were referred from primary care and attended at least one Link Worker session.

## Data collection

We undertook one semistructured interview with each participant between January and June 2016 while they were engaged with Ways to Wellness (length of engagement ranged from 4 months to 14 months). A topic guide was developed covering: referral procedures; level and type of engagement with Ways to Wellness; goal-setting; linkage to other services; long-term condition management; changes resulting from involvement with Ways to Wellness; and views of the service. Interviews took place in participants' homes or an alternative venue of their choosing (often where they attended their Ways to Wellness appointment). Interviews were carried out by two researchers (MS and SL). Following consent procedures, demographic details were collected. Interviews continued until consistencies were identified across participants and data.[16]

## Transcription, data management and analysis

Interviews lasted between 8 min and 1 hour 27 min (average 41 min), were digitally recorded and transcribed verbatim. Field notes were made immediately after each interview and shared among the team. Transcripts were anonymised and checked against recordings for accuracy. Thematic analysis was used[17] with data management supported by NVivo V.10 software.[18] Following close reading of the transcripts, a coding scheme was developed which contained a priori themes based on the topic guide as well as further themes which emerged from the data. The scheme captured data relating to: referral, multimorbidity, experiences of Ways to Wellness service delivery and onward referral, relationship with the Link Worker, impact of Ways to Wellness and barriers to service engagement. The coding framework was applied to an initial randomly selected five interviews, which were double-coded by MS and SL. Following this, the coding frame was reviewed by all team members, modifications agreed and made before being applied to all interviews. Line-by-line coding and constant comparison were used to code the entire data set;[19 20] deviant case analysis, where we sought out opinions which modified or contradicted the analysis, was used to enhance validity.[21]

## RESULTS

The impact of Ways to Wellness is described by our detailed analysis of the following three themes: negative impact of long-term conditions and multimorbidity; Link Worker roles; and positive impact of the programme.

## Participant characteristics

As shown in table 1, 30 adults, 14 women and 16 men, aged 40–74 years (mean age 62 years) took part. Thirteen participants were over state pension age; 4 of the 17 participants of working age were in employment. Occupational social class[22] based on current or previous main employment indicates that the sample included individuals from across the social class spectrum, excepting social class 1; social classes 2–4 accounted for two-thirds of the sample and the remaining third were from social classes 5–8. Five participants were from black and minority ethnic communities, 1 identified as white Irish and the remaining 24 were white British. Multimorbidity was a prominent feature of the sample. Based on self-reported health conditions, only one participant (15) had a single long-term condition. Most participants had more than one 'referral' long-term condition, had other health problems and associated mental health issues, low confidence and social isolation. With the exception of the diagnosed long-term conditions that triggered a referral to Ways to Wellness (ie, diabetes (types 1 and 2), chronic obstructive pulmonary disease, asthma, coronary heart disease, heart failure, epilepsy, osteoporosis) other physical and mental health problems were self-reported at interview. No further assessment of physical or mental health was made during the study.

At the time of interview, participants had been receiving the social prescribing programme for between 4 months and 14 months. Table 1 shows that the number of services that each participant reported being linked into ranged from 0 to 5; the average number was 1.7. However, seven participants (11, 12, 13, 16, 24, 25, 30) obtained welfare benefits advice from their Link Workers, rather than being linked into welfare rights services, demonstrating that some Link Workers drew on their specialist knowledge to assist their client rather than make a referral. Table 2 shows that 54 referrals to community, voluntary and NHS services were made for the 30 participants. These are categorised as follows: long-term condition management; mental health; physical activity; weight management and healthy eating; NHS services (eg, physiotherapy), welfare rights advice (eg, benefits advice, aids and adaptations), learning/employment assistance (eg, curriculum vitae (CV) writing), voluntary work, arts-based activities (eg, choir, art therapy), community-based activities (eg, gardening, fishing, crafts). Services promoting physical activity were the most common linkage.

## Impact of long-term conditions and multimorbidity

All participants had been deeply affected, physically, emotionally and socially by their health problems. Physical effects included pain, sleep problems, side effects of medication, continence issues and significant functional limitations. With increasing age, existing conditions worsened and multimorbidity increased:

**Table 1** Demographic characteristics and long-term health conditions of study participants

| ID | Sex | Employment status | Occupational social class *,†,‡,§ | No. of WtW conditions¶ | Total no. of non-WtW conditions** | Mental health/social isolation†† | Age band (years) | Months involved with WtW at interview | No. of services linked into |
|---|---|---|---|---|---|---|---|---|---|
| 1 | Male | Retired | 7 | 3 | 3 | Yes | 70–74 | 4 | 2 |
| 2 | Female | Retired | 2 | 3 | 3 | Yes | 70–74 | 4 | 3 |
| 3 | Female | Employed | 7 | 1 | 1 | Yes | 45–49 | 5 | 1 |
| 4 | Female | Unemployed | 8 | 2 | 1 | Yes | 55–59 | 5 | 2 |
| 5 | Female | Retired | 2 | 2 | 4 | Yes | 65–69 | 6 | 1 |
| 6 | Male | Retired | 4 | 2 | 1 | Yes | 65–69 | 7 | 0 |
| 7 | Male | Unemployed | 7 | 2 | 1 | No | 55–59 | 7 | 1 |
| 8 | Female | Unemployed | 2 | 2 | 3 | No | 55–59 | 7 | 2 |
| 9 | Male | Employed | 4 | 3 | 3 | Yes | 55–59 | 7 | 5 |
| 10 | Male | Unemployed | 4 | 2 | 4 | Yes | 60–64 | 7 | 4 |
| 11 | Male | Unemployed | 2 | 2 | 4 | Yes | 45–49 | 7 | 5§§ |
| 12 | Male | Unemployed | 4 | 1 | 3 | Yes | 55–59 | 7 | 1 §§ |
| 13 | Male | Unemployed | 7 | 1 | 2 | Yes | 60–64 | 7 | 0§§ |
| 14 | Male | Unemployed | 2 | 2 | 1 | No | 60–64 | 7 | 5 |
| 15 | Female | Retired | 8 | 1 | 0 | No | 70–74 | 8 | 3 |
| 16 | Female | Retired | 2 | 2 | 0 | No | 65–69 | 8 | 0§§ |
| 17 | Female | Unemployed | 3 | 2 | 0 | Yes | 50–54 | 9 | 1 |
| 18 | Female | Retired | 3 | 3 | 2 | Yes | 65–69 | 10 | 1 |
| 19 | Male | Retired | 6 | 2 | 4 | Yes | 65–69 | 10 | 2 |
| 20 | Male | Retired | 4 | 2 | 1 | No | 70–74 | 11 | 1 |
| 21 | Female | Retired | 5 | 3 | 1 | No | 70–74 | 12 | 3 |
| 22 | Female | Unemployed | 3 | 2 | 2 | Yes | 40–44 | 12 | 1 |
| 23 | Male | Retired | 6 | 2 | 4 | No | 70–74 | 12 | 2 |
| 24 | Female | Retired | 2 | 2 | 0 | Yes | 70–74 | 12 | 0 §§ |
| 25 | Female | Unemployed | 2 | 3 | 1 | Yes | 50–54 | 12 | 0 §§ |
| 26 | Male | Retired | 3 | 2 | 3 | No | 70–74 | 13 | 3 |
| 27 | Male | Employed | 4 | 1 | 1 | Yes | 40–44 | 14 | 0 |
| 28 | Male | Retired | 4 | 2 | 1 | No | 70–74 | 14 | 3 |
| 29 | Male | Employed | 6 | 1 | 3 | Yes | 60–64 | NK‡‡ | 1 |
| 30 | Female | Unemployed | 2 | 2 | 2 | Yes | 45–49 | NK‡‡[h] | 1§§ |

Continued

**Table 1** Continued

| ID | Sex | Employment status | Occupational social class *,†,‡,§ | No. of WtW conditions¶ | Total no. of non-WtW conditions**[22] | Mental health/ social isolation†† | Age band (years) | Months involved with WtW at interview | No. of services linked into |
| --- | --- | --- | --- | --- | --- | --- | --- | --- | --- |

*SOC2010 volume 3: the National Statistics Socio-economic classification (NS-SEC rebased on SOC2010).

†Last occupation used except when had changed due to health issues and occupation before health forced change was used for classification.

‡Occupations of NS-Sec analytical class coded using classification tool found at: https://www.ons.gov.uk/methodology/classificationsandstandards/standardoccupationalclassificationsoc2010/soc2010. Last accessed 21 September 2016.

§1 = 2 = Lower managerial, administrative and professional occupation. 3 = Intermediate occupation. 4 = Small employers and own account workers. 5 = Lower supervisory and technical occupation. 6 = Semiroutine occupation. 7 = Routine occupation. 8 = Never worked/long-term unemployed.

¶Conditions which triggered a referral to Ways to Wellness were: diabetes (types 1 and 2), chronic obstructive pulmonary disease, asthma, coronary heart disease, heart failure, epilepsy, osteoporosis) with or without anxiety or depression. This column indicates the number of 'Ways to Wellness' long-term conditions that participants had been diagnosed with by a medical practitioner.

**Based on self-report at interview.

††A broad category that includes low mood, anxiety, depression, loneliness and social isolation and is based on self-report at interview where participants described or reported these conditions or feelings.

‡‡Not known

§§Received welfare benefits advice from Link Worker and were not referred onto specialist welfare rights services.

WtW, Ways to Wellness.

*'I've got COPD, high blood pressure, diabetic … a prolapsed disc and … an operation on it that didn't work, so I'm in constant pain with my back. Just generally not fit, not well … stent put in because I had kidney stones. They were supposed to take it out … but because I had a temperature they didn't take it out. They left it in, but I'm having nothing but trouble with it, water infections, feeling as if I want to go to the toilet all the time and when I do go I don't pass anything. It's like dying for a wee all the time… It's got me really down.'* (P1, male, 70–74 years)

Some of those experiencing depression reported that this had been with them for many years, but most accounts of depression and/or anxiety were attributed to their long-term condition(s) and the difficulties of carrying out a 'normal' life. Wider participation within family and community networks was affected, as well as basic activities of daily living such as shopping or housework. Social isolation was common among many of the participants, but for those of working age, unemployment, lowered incomes and reliance on sickness welfare benefits were commonplace. The overall picture was of lives seriously blighted by long-term health problems:

*'I want to get back to work. I was used to doing things and it is really hard not being able to do all the things I used to do, yes, and I was depressed at the time as well … it looks like a long road to get back to being healthy again; and I needed all the help I could get … I have got asthma and I have got diabetes and high blood pressure. And since I have had the treatment for cancer … I was fine after the operation but it was the radiotherapy that really knocked me out … I mean they do warn you that it affects your energy – but my energy levels just plummeted and I wasn't able to do hardly anything.'* (P8, female, 55–59 years)

### Link Worker roles
#### Connecting with service users
Link Workers operated an accessible, flexible service between 9:00 and 17:00, Monday to Friday. Following primary care referral, Link Workers contacted service users by telephone to arrange an initial appointment, which could be held at GP practices, community centres, cafés, or infrequently, participants' homes. In general, contact with Link Workers was more frequent during the initial stages, reducing as the relationship developed and the service user became more independent; however, contact also decreased or increased with current need. Face-to-face contact was also supplemented by telephone, email or text:

*'[Link Worker] says, 'I'm there, basically, any time,' obviously within working hours, but she says, 'Just phone me up if you need me, and any questions or anything.' So basically I can see her as often as I want or as little as I want, but she likes me to keep her informed of anything happening, so she knows.'* (P11, male, 45–49 years)

**Table 2** Number of onward referrals by service category among interview sample

| Category | Number of referral to other services |
|---|---|
| Long-term condition management (eg, voluntary sector support groups) | 7 |
| Mental health (eg, CBT) | 1 |
| Physical activity (eg, gym, walking group, swimming) | 21 |
| Weight management/healthy eating | 5 |
| NHS services (eg, physiotherapy) | 7 |
| Welfare rights advice (eg, benefits advice, aids and adaptations) | 3 |
| Learning/employment assistance (eg, CV writing) | 5 |
| Voluntary work | 1 |
| Arts-based activities (eg, choir, art therapy) | 2 |
| Community-based activities (eg, gardening, fishing, crafts) | 2 |

CBT, cognitive behavioural therapy.

Participants appreciated the flexibility and 'open door' nature of Ways to Wellness, although this could be limited for those who were working. Of particular value was the potential to be engaged with the service for up to 2 years. Due to the long-term and complex nature of conditions which often fluctuated, participants recognised that a shorter-term approach would be inadequate, particularly when accounting for wider life events such as job loss or bereavement:

*'Well, I think you can dip in and dip out. It's the kind of thing if you need them, you phone them and they'll get straight back to you. They're there, I know they're there … if something happens to me now.'* (P13, male, 60–64 years)

### Link Worker approach

Participants consistently reported feeling at ease and relaxed with their Link Worker, which enabled them to develop an open and trusting relationship. The Link Workers' person-centred and non-judgemental approach facilitated trust, feelings of control and a readiness to reflect on current circumstances and implement positive changes. Service users felt listened to and valued:

*'… [Link Worker] really helped me along … they're not judgemental … and they were so easy going and they were just lovely.'* (P17, female, 50–54 years)

*'She [Link Worker] just brought things up … we'd have a discussion … we just started talking and that's when I just came out with different things … and she gave the feeling that you could open up to her … where some people, you are guarded because you don't want them to know too much of your private life.'* (P9, male, 55–59 years)

A central part of the Link Worker role was to facilitate engagement with other services, as shown in table 2. The level and type of support offered to facilitate engagement varied, and was balanced against service users' need and readiness to engage with other services. Participants appreciated not feeling under undue pressure

and recounted how the right amount of encouragement supported them to engage and make changes at their own pace. Narratives indicated service users experienced a considerable degree of control over their onward referral which helped them feel that when they did engage with other services, they were ready to do so:

*'There's no pressure or anything. If you can't do it, they don't ask why, they just say, 'Oh, we'll try to sort something else out for you,' She said, 'Well, it needs to come from you.' … if it was, like, a sort of forced thing I think I would have problems going [to self-help organisation] … I hate it if I'm made to go.'* (P11, male, 45–49 years)

*'I certainly felt better from having spoken to her [Link Worker], and for her inspiring me to … kick me along and say, 'Yes, you can do that, you can do that.' And when I described the [negative] feelings I had she turned them around to be something else that made me feel loads better.'* (P5, female, 65–69 years)

The level of support that some service users required in order to engage with services, particularly those involving physical activity, appeared to be considerable. Link workers paced the level of support they offered, particularly in the initial stages:

*'I just expected the Link Worker to introduce me to the gym, and that would have been it. And I think, if it had just been [that] I would have turned round, and I would have gone the opposite direction. But because of the way it was so gradually and really professionally linked in to different things, I just felt as though I'd floated into it, rather than getting shoved from behind. I just felt as though I was gradually moved into it.'* (P2, female, 70–74 years)

Building self-confidence, self-reliance and independence was another facet of the Link Workers' approach, managed through ongoing support and persistence in finding the right motivational tools for the individual, while conveying the need for personal responsibility and

resilience. This enabled service users to make changes to their lives, engage with other organisations and manage their long-term conditions:

*'I mean, she's just someone who can make you do things… Not as in a bad way, she sort of, like, empowers you, for want of a better word, to do it, you know what I mean? She gives you, '… give it a go,' and she'll explain… and if you don't go she doesn't get disappointed or anything like that, she just says, 'Oh, right, well, we'll sort something else out for you."… I think the only person that can help is myself, really. I think it's 99% me and the 1% of help I get from the rest of the people, but … they can only like, give you advice and encouragement, the rest has got to be yourself.'* (P11, male, 45–49 years)

The holistic service offered by the Link Workers was contrasted favourably with what was available or possible through the GP or with previous attempts by other health professionals to deal with complex health issues:

*'I couldn't go to my doctors and ask them, I couldn't phone the nurse and ask the nurse.'* (P13, male, 60–64 years)

*'It hasn't worked miracles, but it's nice that she's there and sometimes you don't bother your GP or whatever, you know. It's a brilliant service, I think.'* (P26, male, 70–74 years)

*'At the very beginning, we agreed that she'd work around me, not me around her, to work on the foods that I thought were suitable for me … I met up with dieticians before … I felt they were dictating the food I should have, not the food I wanted to have. So that's what surprised me about Ways to Wellness, they're working around me … it's different. You talk about it. You're relaxed … You're in charge of it.'* (P6, male, 65–69 years)

### Positive impact of the Link Worker social prescribing programme

#### Health-related behaviours

Many participants attended physical activity classes specifically tailored to their health needs, helping them to incrementally build confidence and strength to improve their physical health. Some participants noted that as a result of the Ways to Wellness programme, their level of fitness had improved:

*'I really enjoyed it … When I first started off, I was doing five min on the cycling machine and … I had that up to 20 min … I was up to 15 min on [the cross trainer] and I was pulling weights … they asked us all these relevant questions because of my medical history so I hadn't to do too much for my heart … I just feel a hell of a lot fitter.'* (P28, male, 70–74 years)

Participants who received healthy eating advice and cooking skills recounted in some detail how they had reduced their intake of unhealthy food and how they were equipped with the practical skills to incorporate healthier foods into their diet:

*Certainly the [name of course], because that was a six week course…The first session was salad and pizza, then we did stir fry, soup last week. Next week it's going to be fish, then I think it's desserts on the sixth week. Brilliant. It's not boring, but I just can't believe how good I've found it. It's not just for cooking. … it's what you should be eating, it's the portion size which is so important.'* (P14, male, 60–64 years)

*'I was asked to do a food diary. You see straightaway that what you're doing is wrong. I think that was one of the biggest things why I lost quite a bit of weight, to be honest … you don't realise at the time, but it was nothing for me to eat eight slices of bread a day. That's where all the badness is. Now, I only have pitta bread, really, or a small brown sliced bread. That was a big help, with the directions pointing to the food … it's smaller portions and more variety as well … now, I eat more fruit in the mornings and in the afternoons, and I don't pick as much.'* (P29, male, aged 60–64 years)

Participants described how these diet and physical activity lifestyle changes improved their physical health. Many noted weight reduction and increased energy and fitness that enabled them to break a cycle of illness and restart activities, such as walking, shopping and gardening, which previously they felt unable to manage:

*'At one time, when I went up [a hill]… I used to stop … but now I can walk directly up…  I can carry my shopping bags now … I'd get up and dance … [without feeling] I needed to sit down because I was out of breath.* (P2, female, 70–74 years)

#### Mental health

Social isolation, low mood, anxiety and depression were commonly experienced. For some, this was related primarily to physical health, but for others, this was compounded by life events such as bereavement. Offering opportunities for activities, which allowed people to meet and socialise in the community, reduced social isolation and positively impacted on self-confidence, self-esteem and mental well-being:

*'After [partner] passed away I was, not a recluse, but I just didn't want to talk to anybody. But since I've been coming to see [the Link Worker] I've broadened my horizons and I get out … I've got a lot more confidence.'* (P10, male, 60–64 years)

*'Before the Ways to Wellness, I used to just stay in. But they explained that if you get out and about, it helps you, which has proven right … there were times when I wasn't feeling too good, and I thought, 'I can't be bothered,' but I've pushed myself. By the time I've got out and got back, I've felt 100% better … my health has changed a lot … it's made me feel a lot more confident that there's somebody there.'* (P4, female, 55–59 years)

Long-term health problems were linked to a range of social and economic factors including relationship breakdown, job loss, income reduction, debt and housing problems. Link Worker consultations noted socioeconomic factors

and, where possible, took practical action to maximise income, reduce debt and assist service users navigating the welfare benefits system:

*'Whatever money I owed like electricity and TV licence was in my mind always eating me from inside. I sorted out that and it just changed so many things … It changed my attitude, it changed my behaviour and it changed my mood … I am not depressed like before … I feel better about everything … I go out almost every day … I play more, I write, I do jobs at home … I read more … yes, a big change.'* (P30, female, 45–49 years)

*'Because of my language barrier some things I could not sort things out so easily, but she could. Even though I didn't know about Attendance Allowance she applied for that … I didn't know which benefit I could get. She said, 'I can try this one.' She tried it and she was successful and it helped me a lot.'* (P16, female, 65–69 years)

For those of working age, health-related unemployment was a major problem. Steps to assist with finding paid or volunteer work, returning people to work or having reasonable adjustments to work settings were undertaken and highly appreciated:

*'I felt 100% better after talking to [Link Worker], she was just excellent … She put us in contact with people [to] do a new CV and look for a new job, [and] found information out for us … which has given me the confidence to go back to my company … things were put in place at work, so that I could take my breaks when I needed them … It made a massive difference to me, personally … I was getting on board with [mental health condition], and now, I'm at work with renewed vigour.'* (P9, male, 55–59 years)

### Long-term condition management

Weight reduction and increased fitness helped participants manage the pain and tiredness experienced as a result of their long-term health conditions. Some participants with diabetes noted being in better control of their condition due to reduced cholesterol and sugar levels. Accounts of positive improvements often followed years of worsening health and poor long-term condition management:

*'The instructor here has given me a programme that I'm working on to help build up the muscles for the legs and for my back … I'm managing [my arthritis] with the help of the gym … [also] my sugar levels … [have] come down to 4.9 … [and] my cholesterol was 3.1, which is good … all that had to do with Ways to Wellness and the exercise I've been doing.'* (P6, male, 65–69 years)

Long-term condition management services that service users were directed to by Link Workers were highlighted as extremely helpful, particularly the combination of expert and peer-led advice on coping and symptom management strategies:

*'The [name] group was wonderful because there were people there who had similar conditions … and they have medical*

*professionals come in to talk about things that are specific to people with chest complaints … coping mechanisms … gentle exercises and things to help. It improves your stamina, your breathing abilities and everything. So yes, that has been absolutely wonderful … honestly it was the turning point in my life. Because I had reached rock bottom and I just needed practical help to try and get back up … you have to manage your expectations as well as your illness and it has helped considerably.'* (P8, female, 55–59 years)

Regularly engaging with services was challenging, particularly for people whose condition fluctuated and those suffering from more than one health problem. Service users worried about not always being able to attend, which was often a reason for not sustaining engagement with services in the past:

*'I was really down, so I couldn't go on Friday. It's been like that since I first joined. I've been missing [sessions] because of my COPD … bad turns … chest infections … [but] they [Ways to Wellness] restarted me again, so they've looked after me … they've been really, really good … they've stuck with me.'* (P1, male, 70–74 years)

Participants experienced Ways to Wellness as an approach which recognised the long-term and complex nature of their health problems. It supported realistic, progressive and personalised goal-setting. Participants' expectations of progress were therefore achievable and reflected that a long-term approach was necessary to make improvements, helping people to live with their conditions and improve their well-being:

*'I am on the road but it is slow … work in progress … if it was easy I would have done it years ago … I have been well impressed [with Ways to Wellness]… because they have a very practical approach and know that it has got to be incremental … you can't do everything at once so you have got to start small and build up … they have got the big picture in mind but lots of little steps in between to get there … I would like to get back to being wonderful in the next five min it is not going to happen. So I know that I have got to take it bit by bit so it sticks.'* (P8, female, 55–59 years)

*'[Link Worker] epitomises the word, wellbeing, because that's what she's got in mind for you, your wellbeing. Nothing else, she gives the impression that you're the only one that she's looking after, at that time, in that moment in time, you're the only one that matters to her.'* (P9, male, 55–59 years)

### Discussion

This study provides evidence about the impact of a Link Worker social prescribing programme on health and well-being via improvements in health-related behaviours and long-term condition management. The specific model of social prescribing examined here comprised personalised support together with linkage into appropriate services facilitated by a Link Worker. The data reveal that key elements of the model are that it is: long-term in nature; addresses the coexistence of multimorbidity, mental health problems and social isolation; and, where

applicable, tackles related socioeconomic issues. This study demonstrates that the rapport and quality of the relationship between the Link Worker and service user was central to achieving well-being, as well as key to successfully linking service users into a wide range of community, voluntary and NHS services identified as relevant to their situation. Change in health-related behaviour and long-term condition management was facilitated through the use of setting realistic, progressive and personalised goals, problem-solving, receiving regular feedback and social support. These behaviour change techniques have been shown to be effective in other lifestyle interventions.[23–25] Crucially these techniques were adopted while also supporting individuals to address social and economic problems. The Link Worker social prescribing programme engendered feelings of control and self-confidence, reduced social isolation and led to positive physical changes such as weight loss, increased physical activity, improved long-term condition management and mental health, greater resilience and effective coping strategies to manage relapses.

Psychosocial problems, particularly mental health conditions and social isolation, are the most common reasons for referral into social prescribing programmes.[4 6] A review exploring the social prescribing evidence base identified 24 studies diverse in methodology and the service being evaluated.[6] The quantitative evidence in the review was limited with the exception of one randomised controlled trial demonstrating clinically important benefits in managing psychosocial needs at 4 months.[26] The quality of eight studies using quantitative methods was poor, with small samples and high dropout rates.[6] Qualitative evidence, from this review and more broadly, also varies in quality, but generally indicates that those referred into social prescribing schemes report improved mental well-being and reduced social isolation.[6 9 27] A systematic review and meta-analysis of exercise referral schemes, a type of social prescribing intervention, targeted at sedentary individuals with or without a medical diagnosis was inconclusive with regards to increasing physical activity, although the heterogeneity of the quality and the nature of the schemes is likely to have contributed to this conclusion.[28] The data presented in our study provide a much more in-depth account of *how* Link Worker social prescribing positively influences long-term condition management and how this programme can lead to improved physical and mental health outcomes.

### Strengths, limitations and implications

The strength of this work lies in the depth and consistency of the service user accounts regarding the impact of Link Worker social prescribing. Methodologically, we followed accepted practice in fieldwork, analysis and interpretation.[29 30] There are a number of limitations. First, although the sample did include people with a wide range of long-term conditions, almost all with multimorbidity, and many experiencing social isolation and mental health problems we did only interview people who agreed to a referral, then enrolled into and remained with Ways to Wellness. We therefore cannot make claims about patients with long-term conditions who did not engage in the first place or who dropped out of the programme early. Second, we cannot be certain of the extent to which the experiences of the 30 study participants reflect those of the 1603 who used the service, although maximum variation sampling was used to include a wide range of participant experience. Third, the study lacks precise data about levels of participation in 'linked' services. We therefore cannot draw specific conclusions about the effects on health and well-being with respect to intensity and duration of 'linked' service use. These areas are undoubtedly important for further research.

Long-term health conditions are arguably the greatest challenge facing the NHS; they have an adverse effect on quality of life and reduce life expectancy. Multimorbidity accounts for over 70% of total healthcare spend.[31] Non-medical interventions are proposed as a cost-effective alternative to foster self-management in people with long-term conditions.[32 33] People from lower socioeconomic groups experience higher levels of chronic disease, and also have poorer condition management, worse health outcomes and higher mortality.[31] Behavioural risk factors for long-term conditions include poor diet, smoking and physical inactivity, all of which are socioeconomically patterned.[34] However, lifestyle behaviours are also affected by wider health determinants and are not simply choices that individuals make.[35 36] Supporting people to make meaningful and sustainable lifestyle changes is complex, time-consuming, and a challenge for time-limited GPs.[11] Given the complex and long-term nature of the problems faced by service users, it is unsurprising that a holistic and relatively intensive approach, as offered by Ways to Wellness, is required in order to facilitate and maintain behaviour change, and it is clearly not possible, nor appropriate to offer this level of support routinely in primary care. Referral into social prescribing programmes is likely to continue due to the need to tackle long-term health conditions, multimorbidity and an ageing population in the face of stretched NHS resources.[33 37] The range of approaches to, and models of, social prescribing[38] make it challenging to amass good quality evidence to inform commissioning.[37 38] In order to inform practitioners and commissioners, future research should be robust and comparative, addressing when, for whom and how well the programme works, including validated measures of intervention impact and accurate costs.[37] To this we would add further qualitative, particularly ethnographic, research to examine potential wider impacts on families and communities and how community-based, non-medical and personalised interventions supplement healthcare.

**Acknowledgements** The authors thank the participants for sharing their experiences, Link Workers and Link Worker Managers at Mental Health Concern, First Contact Clinical, Healthworks and Changing Lives for their assistance with recruitment, and Chris Drinkwater, Tara Case and Amie Callan from Ways to Wellness and Dr Guy Pilkington from Newcastle Gateshead Clinical Commissioning Group for their support.

**Contributors** SM, LP and NO'B conceived and designed the research and obtained funding. MS and SL undertook interviews. SM, MS, LP and SL contributed to the data analysis. SM, MS, SL, LP and NO'B wrote the paper. All authors read and approved the final version.

**Funding** This work was funded by the Cabinet Office of the UK Government (grant number A1543). The researchers are independent of the funders and the views and opinions expressed in this paper are those of the authors and not necessarily those of the Cabinet Office. The funders and sponsors (Newcastle University) had no role in the design, collection, analysis and interpretation of the data, or the writing of this paper or the decision to submit this article for publications. All authors had full access to the data in the study and can take full responsibility for the integrity of the data and the accuracy of the analysis. Suzanne Moffatt is a member of Fuse, the Centre for Translational Research in Public Health. Fuse is a UK Clinical Research Collaboration (UKCRC) Public Health Research Centre of Excellence. Funding for Fuse from the British Heart Foundation, Cancer Research UK, Economic and Social Research Council, Medical Research Council, the National Institute for Health Research, under the auspices of the UKCRC, is gratefully acknowledged. The views expressed in this paper do not necessarily represent those of the funders or UKCRC. The funders had no role in study design, data collection and analysis, decision to publish, or preparation of the manuscript.

**Competing interests** None declared.

**Patient consent** Detail has been removed from this case description/these case descriptions to ensure anonymity. The editors and reviewers have seen the detailed information available and are satisfied that the information backs up the case the authors are making.

**Ethics approval** Newcastle University Faculty of Medical Sciences Research Ethics Committee.

**Provenance and peer review** Not commissioned; externally peer reviewed.

**Data sharing statement** No additional data are available.

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
