## [Reviewer comments · BMJ Open]

ARTICLE DETAILS

TITLE (PROVISIONAL)	Link worker social prescribing to improve health and wellbeing for people with long-term conditions: qualitative study of service user perceptions
AUTHORS	Moffatt, Suzanne; Steer, Mel; Lawson, Sarah; Penn, Linda; O'Brien, Nicola

VERSION 1 - REVIEW

REVIEWER	Helen Chatterjee University College London, UK
REVIEW RETURNED	08-Dec-2016

GENERAL COMMENTS	This study is valuable as it provides evidence for the impact of link worker schemes on health and wellbeing among adults with long-term conditions living in an area of high socio-economic deprivation, but the study does not appear to be evaluating a social prescribing intervention. The Abstract does not state what the intervention, simply referring to a Social Prescribing intervention. The term social prescribing is a catch all term which includes a range of diverse activities from arts and creative activities, to exercise and green therapies. The Abstract should be clearer about what the activities were and frequency of participation (i.e. dosage). The Introduction is fairly robust but lacks a comprehensive description of SP; it would be helpful for the authors to include their preferred definition of SP and examples, so that readers who are not familiar with the term get a sense of the scope of interventions which come under the SP umbrella. The study population is adequately described and good justification is made for the for the target cohort. There is however insufficient information about a) the nature, duration and frequency of the interventions under study and b) the data collection methodology. Regarding (a) much more information is needed about the social prescribing interventions as it is not clear what these activities were or how frequently participants (ps) engaged in them; without these data it is impossible to gauge the efficacy of the interventions or make recommendations about future implementation. The authors mention a range of community services including welfare rights advice, walking groups, physical activity classes, arts groups, continuing education, but it is not clear which of these were undertaken by the study cohort. Details such as duration and frequency of the intervention are critical to understand their value
---

	and impact. Further details are also required regarding referral mechanisms – again this is a core area which needs further study so it is a shame that this is not discussed here, or indeed feedback mechanisms between the SP activities and the link workers. Regarding (b) further details are needed about the data collection protocol; for example, how many interviews were undertaken with each participant and at what stage in the programme (at the end?)? Clarity is also needed regarding the number of individuals included in the thematic analysis; 30 participants are reported at the start and the authors state that they applied a coding framework to five participants; what is the rationale for this and how were these 5 ps chosen? The results are interesting in understanding the impact of long term health conditions on ps from low socio-economic status backgrounds, many of whom report social isolation. The study also adds value to understanding the possible benefits of link worker programmes, but critically the study does not add anything to the debate or evidence base regarding social prescribing. This paper purports to be assessing the impact of a SP intervention but in fact appears to be evaluating the efficacy of a link workers programme; the title and main thrust of the paper are misleading in this context and without further information it would appear that it is not in fact a social prescribing programme but instead a link/navigation programme that includes (amongst other things) linking with community based activities (which may be social prescribing interventions but without further details this is not possible to ascertain). In this context the paper needs to be redrafted with a new title and further details provided about the activities undertaken and background information on link worker schemes and the links with SP schemes.
--	--

REVIEWER	Richard Kimberlee University of the West of England
REVIEW RETURNED	09-Dec-2016

GENERAL COMMENTS	This is an interesting paper on a very important issue. It adds to the growing literature on social prescribing and usefully provides an in depth insight into the benefits patients experience from engagement with such projects. Importantly, it highlights the importance of the link worker in helping to co-produce improved well-being. It also specifically concentrates on how social prescribing type initiatives can help with long-term condition management. The fact that it is a qualitative study allows us to understand the importance of story in trying to capture the range of benefits that a patient may get. So it deserves to be published and there will be an interested audience to hear its reported insights. However I feel it would benefit from some clarifications and refinements in presentation. Firstly in the text it is reported that most participants experienced multi-morbidity combined with mental health problems, low self-confidence and social isolation; and all were adversely affected physically, emotionally and socially by their health problems. It is not clear in the main body that this is based on self report. This is only mentioned in footnote 7 of the table. I feel it would be better if this is clarified in the article. In particular self-report to who: the GP or to the researchers in interview or to someone else the link worker? I
--

	also think that it needs to be clarified that no tool was used to assess physical or mental health if that was the case. Secondly, I disagree that there is limited evidence about social prescribing impacting on physical health or long-term conditions. In another place they actually say there are no studies, which is not correct. The work in Rotherham reveals some evidence (Dayson and Bashir, 2014) and the Wellspring study (Kimberlee et al, 2014) has quantitative evidence of impact in terms of GP attendance and improved wellbeing on a validated tool. Both studies have long term follow up. The latter has been written up for a peer review article (Kimberlee, 2016). We are not sure how long after referral these patients were interviewed? And additionally the emphasis here is slightly awkward for the methodological approach adopted in the article. The authors seem to lament that there has only been one randomised controlled trial demonstrating impact? The Grant (2000) study was quite small and was a different model to the model outlined here. Should there be more trials? Isn't the strength of undertaking a qualitative study the fact that social prescribing may not be easily assessed using a RCT because there are too many models and the degree of engagement is not always holistic, but often less than that. Thus a trial may not necessarily be appropriate or conducive. Epistemologically the lament doesn't fit with the method deemed appropriate here. Thirdly, Ways to Wellness is defined as the first UK organisation to deliver social prescribing at scale. I think we need some clarification as to what is meant by scale. Thus over what time period is the target of 11,000 going to be reached. Is this a sufficiently large scale? In Gloucestershire the CCG have commissioned social prescribing across the county involving 60 + GP practices and seen over 2,400 plus patients. It has made referrals to over 200 different organisations and has been going since pilots in 2014. The term scale is used throughout the article....so clarification of what the authors mean by a (sufficient) scale would be useful. Finally clarification of the sampling method would be useful. The authors state that they used purposive sampling across the four provider organisations based on the following criteria: age, gender; long-term condition; marital status, employment status, socio-economic status, and level of engagement with Ways to Wellness (intensive to non-intensive). But we are not sure of the actual mix. Is it representative of users of the user or not? And do we know? There appears to be an under representation of employed people compared to others. But we cannot make a judgment on that without more information. There are also various typos eg sometimes using numerals and then words to express value.
--	---

REVIEWER	Jude Stansfield Public Health England England, UK
REVIEW RETURNED	15-Dec-2016

GENERAL COMMENTS	Excellent paper with minor comments/ amends: Background - first line, definition of SP is perhaps too narrow - 'lifestyle' often denotes health lifestyle behaviours but SP is beneficial for addressing causal factors, social isolation, poor mental health. Re 'long term conditions' SP might be beneficial for people with an episode of illness e.g. depression Methods - be useful to say how many (of the eligible ~24000 people
---

	with LTC) are referred to W2W, how many they see in the timescale, and possible of these how many were referred to the research. Table 1 - I think this has too much personal detail in e.g. link workers reading the paper will be able to identify their clients quotes; suggest grouping the data; Funding - it states the sponsor (Newcastle University) were not involved in the research, not sure if this is an error as they conducted the research.
--	---

VERSION 1 – AUTHOR RESPONSE

Reviewer: 1

Reviewer Name: Helen Chatterjee

Institution and Country: University College London, UK

Please state any competing interests or state 'None declared': None declared

Please leave your comments for the authors below

This study is valuable as it provides evidence for the impact of link worker schemes on health and wellbeing among adults with long-term conditions living in an area of high socio-economic deprivation, but the study does not appear to be evaluating a social prescribing intervention.

1.The Abstract does not state what the intervention, simply referring to a Social Prescribing intervention. The term social prescribing is a catch all term which includes a range of diverse activities from arts and creative activities, to exercise and green therapies. The Abstract should be clearer about what the activities were and frequency of participation (i.e. dosage).

RESPONSE: The intervention is now described as a 'Link Worker social prescribing intervention'. We state in the abstract that the intervention is personalised and comprises personalised support to identify meaningful health and wellness goals and linkage into appropriate community services. Unfortunately the word count does not allows us to list the community activities, but we make this clear on p 10, para 2, where we list the types of services that participants were linked into.

2.The Introduction is fairly robust but lacks a comprehensive description of SP; it would be helpful for the authors to include their preferred definition of SP and examples, so that readers who are not familiar with the term get a sense of the scope of interventions which come under the SP umbrella.

RESPONSE: We thank the reviewer for this comment. In the opening paragraph of the paper, we have expanded our description of social prescribing (p5 para 1), as follows, "Social prescribing enables health care practitioners to refer patients to a range of non-clinical services.1 Primarily, but not solely, directed at people with long term conditions, social prescribing harnesses assets within the voluntary and community sector to improve and encourage self-care and facilitate health-creating communities. 2 3 4 " We have re-ordered the introduction and extended the paragraph on definitions, explanations and included our preferred definition of social prescribing (SP), as follows (p5-6), "Services into which patients are referred vary, and can include activities that involve physical activity such as gyms, walking groups, gardening or dance clubs; weight management and healthy eating activities, such as cooking clubs. Addressing wider economic and social issues can involve referral into services which address welfare, debt, housing and employment issues. Groups, such as those targeted at people with specific long term conditions, for example diabetes, chronic obstructive pulmonary disease, may also be accessed via social prescription. Our definition concurs with that of the social prescribing network of Ireland and Great Britain, "enabling healthcare professionals to refer patients to a link worker, to co-design a non-clinical social prescription to improve their health and wellbeing" and use services provided by the voluntary and community sector. (p19).

The study population is adequately described and good justification is made for the for the target

cohort. There is however insufficient information about a) the nature, duration and frequency of the interventions under study and b) the data collection methodology.

3a. Regarding (a) much more information is needed about the social prescribing interventions as it is not clear what these activities were or how frequently participants (ps) engaged in them; without these data it is impossible to gauge the efficacy of the interventions or make recommendations about future implementation.

The authors mention a range of community services including welfare rights advice, walking groups, physical activity classes, arts groups, continuing education, but it is not clear which of these were undertaken by the study cohort. Details such as duration and frequency of the intervention are critical to understand their value and impact. Further details are also required regarding referral mechanisms – again this is a core area which needs further study so it is a shame that this is not discussed here, or indeed feedback mechanisms between the SP activities and the link workers.

RESPONSE: On P6-7 we had detailed (a-e) what the SP intervention in this study comprises. We have added a further section to provide more detail about this highly personalised intervention, P7 “Patient engagement with the intervention varies in terms of intensity, duration, goal setting and onward referral. Patients can remain with the intervention for up to two years; frequency of contact with their Link Worker is mutually agreed, varies depending on need and circumstances, and can be face to face, via telephone, email or text. Data for this study was collected in the first year of intervention implementation. Altogether, the four provider organisations have linked the entire patient population into 133 community and voluntary sector organisations which can be grouped as follows: long term condition management; mental health; drug and alcohol; physical activity; emergency debt, housing hardship; welfare rights advice; housing; volunteering/learning/employment; community activities including gardening, cooking, art, crafts.”

We did not have access to data management systems that provided data on frequency of intervention in terms of Link Worker contact. Frequency of contact was mutually agreed between the service user and the Link Worker as the intervention was designed to be entirely person-centred. On p10, para 2, we have added text to describe the range of services that the participants in this study were referred into, as follows: “At the time of interview, participants had been receiving the intervention for between four and fourteen months. Services that participants reported being linked into were as follows: physical activity/fitness, weight management, healthy eating, long term condition management (i.e. support groups specifically aimed at assisting people manage their long term conditions), welfare rights services (including managing debt, applying for welfare benefits, and/or emergency hardship funds), housing advice, employment advice services, community activities (e.g. choir, swimming, fishing, photography) and other voluntary organisations (e.g. Age UK).”

We explain on P6 para 2, that the referral mechanism into the intervention is, “Following referral from a primary care practitioner (GP, practice nurse, health care assistant)”. On P15, in the results section headed, “Link Worker approach”, we demonstrate how the Link Workers “facilitated engagement with other services” and demonstrate the nuanced, sensitive and person-centred way in which this was done. We also point out in the last paragraph of this section that in making these onward referrals, Link Workers “paced the level of support they offered, particularly in the initial stages”.

Feedback mechanisms between the SP activities and the link workers, whilst important, is beyond the scope of this study.

3b.Regarding (b) further details are needed about the data collection protocol; for example, how many interviews were undertaken with each participant and at what stage in the programme (at the end)? Clarity is also needed regarding the number of individuals included in the thematic analysis; 30 participants are reported at the start and the authors state that they applied a coding framework to five participants; what is the rationale for this and how were these 5 ps chosen?

RESPONSE: We clarify in the 'Data collection' section (P8) that, "We undertook one semi-structured interview with each participant between January and June 2016 whilst they were engaged with Ways to Wellness (length of engagement ranged from 4-14 months)".

All transcripts were included in the thematic analysis, but five transcripts were selected at random for double coding by two members of the research team to check the veracity of the coding framework and to make modifications where required before being applied to the entire data set. We clarify this as follows in the section on P9 headed "Transcription, data management and analysis" and state, "The coding framework was applied to an initial randomly selected five interviews, which were double-coded by MS and SL. Following this, the coding frame was reviewed by all team members, modifications agreed and made before being applied to all interviews."

4. The results are interesting in understanding the impact of long term health conditions on ps from low socio-economic status backgrounds, many of whom report social isolation. The study also adds value to understanding the possible benefits of link worker programmes, but critically the study does not add anything to the debate or evidence base regarding social prescribing.

This paper purports to be assessing the impact of a SP intervention but in fact appears to be evaluating the efficacy of a link workers programme; the title and main thrust of the paper are misleading in this context and without further information it would appear that it is not in fact a social prescribing programme but instead a link/navigation programme that includes (amongst other things) linking with community based activities (which may be social prescribing interventions but without further details this is not possible to ascertain). In this context the paper needs to be redrafted with a new title and further details provided about the activities undertaken and background information on link worker schemes and the links with SP schemes.

RESPONSE: We thank the reviewer for this comment. We have revised the introduction in line with this and the reviewer's first comment (see point 1 above) and added a more complete description of SP and discussed how there are many models of SP. We also make clear in the introduction that there is no one agreed definition of social prescribing. The Social Prescribing Network's short definition of SP, which we quote on P6, "enabling healthcare professionals to refer patients to a link worker, to co-design a non-clinical social prescription to improve their health and wellbeing" and use services provided by the voluntary and community sector. 2 " describes the Ways to Wellness intervention. We therefore feel that it is accurate to describe the Ways to Wellness intervention that we studied a social prescribing intervention, rather than a link/navigation programme. However, in the interests of clarity, we have amended the title of the paper to, "Link Worker social prescribing to improve health and wellbeing for people with long-term conditions: qualitative study of service user perceptions."

Reviewer: 2

Reviewer Name: Richard Kimberlee

Institution and Country: University of the West of England

Please state any competing interests or state 'None declared': None declared

Please leave your comments for the authors below

This is an interesting paper on a very important issue. It adds to the growing literature on social prescribing and usefully provides an in depth insight into the benefits patients experience from engagement with such projects. Importantly, it highlights the importance of the link worker in helping to co-produce improved well-being. It also specifically concentrates on how social prescribing type initiatives can help with long-term condition management. The fact that it is a qualitative study allows us to understand the importance of story in trying to capture the range of benefits that a patient may get. So it deserves to be published and there will be an interested audience to hear its reported

insights. However I feel it would benefit from some clarifications and refinements in presentation.

1. Firstly in the text it is reported that most participants experienced multi-morbidity combined with mental health problems, low self-confidence and social isolation; and all were adversely affected physically, emotionally and socially by their health problems. It is not clear in the main body that this is based on self report. This is only mentioned in footnote 7 of the table. I feel it would be better if this is clarified in the article. In particular self-report to who: the GP or to the researchers in interview or to someone else the link worker? I also think that it needs to be clarified that no tool was used to assess physical or mental health if that was the case.

RESPONSE: We clarify this in three areas of the paper. First, in the section 'Recruitment and Sampling', we add the sentence (P8), "For the purposes of sampling, long-term condition was ascertained by the Link Worker via the referral form, and had therefore been diagnosed by a GP." Second, in the section on 'Participant characteristics' (P10) we state, "With the exception of the long-term conditions which precipitated a referral to Ways to Wellness which were diagnosed by a medical practitioner (diabetes (type 1&2), chronic obstructive pulmonary disease, asthma, coronary heart disease, heart failure, epilepsy, osteoporosis), other physical health and mental health problems were self reported at interview, and no tool was used to assess physical or mental health during the study. Third, on Table 1, we have expanded footnote 5 as follows (P12), "Conditions which triggered a referral to Ways to Wellness were: diabetes (type 1&2), chronic obstructive pulmonary disease, asthma, coronary heart disease, heart failure, epilepsy, osteoporosis) with or without anxiety or depression. This column indicates the number of 'Ways to Wellness' long term conditions that participants have been diagnosed with by a medical practitioner". We then make it clear in footnotes 6 and 7 that other long term conditions and mental health problems were all based on self report at interview.

2. Secondly, I disagree that there is limited evidence about social prescribing impacting on physical health or long-term conditions. In another place they actually say there are no studies, which is not correct. The work in Rotherham reveals some evidence (Dayson and Bashir, 2014) and the Wellspring study (Kimberlee et al, 2014) has quantitative evidence of impact in terms of GP attendance and improved wellbeing on a validated tool. Both studies have long term follow up. The latter has been written up for a peer review article (Kimberlee, 2016). And additionally the emphasis here is slightly awkward for the methodological approach adopted in the article. The authors seem to lament that there has only been one randomised controlled trial demonstrating impact? The Grant (2000) study was quite small and was a different model to the model outlined here. Should there be more trials? Isn't the strength of undertaking a qualitative study the fact that social prescribing may not be easily assessed using a RCT because there are too many models and the degree of engagement is not always holistic, but often less than that. Thus a trial may not necessarily be appropriate or conducive. Epistemologically the lament doesn't fit with the method deemed appropriate here.

RESPONSE: We thank the reviewer for drawing this to our attention. In the introduction, we indicate that most of the research has focused on the impact of SP on mental health/wellbeing, we now include the references on physical health and resource use as follows, (P1) "Whilst less attention has been paid to the impact of social prescribing on physical health and resource use, (Mossabir et al, 2015) improvements in physical activity, (Kimberlee, 2016) reductions in hospital resource use, (Dayson & Bashir 2014) and GP attendance (Kimberlee et al 2014) have been attributed to social prescribing, although longer-term studies with larger sample sizes are required for definitive evidence."

In the discussion, we have deleted the statement (P20) "but provide very little evidence about physical health or long term condition management". However, we feel that it would be an omission not to include the very brief reference that we make to the Grant et al (2000) RCT as it is mentioned in the reviews we discuss. We do end this section of the discussion by pointing out the value of this qualitative study in providing a much needed in-depth account of how social prescribing positively

influences long term condition management. In the final paragraph of the discussion, we point out that the different models of social prescribing makes it challenging to amass a robust evidence base. In addition to quantitative evidence, we also call for more in-depth qualitative and particularly ethnographic studies to fully investigate the potential and impact of SP.

We are not sure how long after referral these patients were interviewed?

RESPONSE: This information has been added to Table 1, P11, and indicates that participants were interviewed between four and 14 months after their referral to Ways to Wellness.

3.Thirdly, Ways to Wellness is defined as the first UK organisation to deliver social prescribing at scale. I think we need some clarification as to what is meant by scale. Thus over what time period is the target of 11,000 going to be reached. Is this a sufficiently large scale? In Gloucestershire the CCG have commissioned social prescribing across the county involving 60 + GP practices and seen over 2,400 plus patients. It has made referrals to over 200 different organisations and has been going since pilots in 2014. The term scale is used throughout the article....so clarification of what the authors mean by a (sufficient) scale would be useful.

RESPONSE: We have modified the text to indicate that Ways to Wellness is among the first organisations to deliver SP on a large scale, and we state that the time period to reach the target of 11,000 users is over seven years, (P6) "Ways to Wellness 12 is one of the first UK organisations to deliver social prescribing on a large and prolonged scale; funded for seven years through a social impact bond model, with an overall target of 11,000 users over this period." Further, on P7 we point out that Ways to Wellness have 'linked' service users into 133 different voluntary and community organisations and we outline the categories of these organisations.

4.Finally clarification of the sampling method would be useful. The authors state that they used purposive sampling across the four provider organisations based on the following criteria: age, gender; long-term condition; marital status, employment status, socio-economic status, and level of engagement with Ways to Wellness (intensive to non-intensive). But we are not sure of the actual mix. Is it representative of users of the user or not? And do we know? There appears to be an under representation of employed people compared to others. But we cannot make a judgment on that without more information.

RESPONSE: The purposive sampling strategy was intended to produce a sample of maximal variation on the basis of the criteria selected, i.e. age, gender; long-term condition; marital status, employment status, socio-economic status, and level of engagement with Ways to Wellness (intensive to non-intensive). The resulting sample varied according to each of these criteria. However, due to data protection and restrictions in our access to data, it is not possible to compare the characteristics of this qualitative sample with the larger group of Ways to Wellness service users during the period of interview. We have presented the data to Link Workers delivering the intervention who have verified the considerable degree and impact of multi-morbidity, mental health/social isolation amongst the service users that we identified in this sample is common among Ways to Wellness service users.

There are also various typos eg sometimes using numerals and then words to express value.

RESPONSE: The paper has been carefully proof read, typos removed and consistency in expression of value

Reviewer: 3

Reviewer Name: Jude Stansfield

Institution and Country: Public Health England, England, UK

Please state any competing interests or state 'None declared': None

Please leave your comments for the authors below

Excellent paper with minor comments/ amends:

1. Background - first line, definition of SP is perhaps too narrow - 'lifestyle' often denotes health lifestyle behaviours but SP is beneficial for addressing causal factors, social isolation, poor mental health. Re 'long term conditions' SP might be beneficial for people with an episode of illness e.g. depression

RESPONSE: We have taken on board this helpful comment and have amended the opening sentences to read as follows (p5, para 1), "Social prescribing enables health care practitioners to refer patients to a range of non-clinical services. 1 Primarily, but not solely, directed at people with long term conditions, social prescribing harnesses assets within the voluntary and community sector to improve and encourage self-care and facilitate health-creating communities.

2. Methods - be useful to say how many (of the eligible ~24000 people with LTC) are referred to W2W, how many they see in the timescale, and possible of these how many were referred to the research.

RESPONSE: We have stated on P6, para 2 that over the seven year funding period the overall target is for 11,000 people with long term conditions to be referred and seen by Ways to Wellness. We provide the sampling and recruitment method on P8, and explain that the Link Workers acted as gatekeepers for researcher access and helped identify potential participants on the basis of the following criteria; age, gender; long-term condition; marital status, employment status, socio-economic status, and level of engagement with Ways to Wellness (intensive to non-intensive). We also state that we continued sampling and interviewing until data saturation was reached (P9, para 1).

3. Table 1 - I think this has too much personal detail in e.g. link workers reading the paper will be able to identify their clients quotes; suggest grouping the data;

RESPONSE: We have removed age and included age range in five year bands to preserve anonymity

4. Funding - it states the sponsor (Newcastle University) were not involved in the research, not sure if this is an error as they conducted the research.

RESPONSE: This refers to research governance and sponsorship of the research. Newcastle University sponsorship was entirely separate from the research team who conducted the research

VERSION 2 – REVIEW

REVIEWER	Helen Chatterjee University College London, UK
REVIEW RETURNED	01-Feb-2017

GENERAL COMMENTS	The MS is improved now that authors have clarified that they are evaluating the Link Worker referral programme and not the actual Social Prescribing interventions (i.e. arts activities, volunteering, etc.) though I still do have concerns there is some confusion here. It would help if the Objectives section in the Abstract is clarified; for example it currently states that: Objectives: To describe the experiences of patients with long-term conditions who are referred to and engage with a Link Worker social prescribing intervention and identify the impact of the intervention on their health and wellbeing. The study is not evaluating the impact of the SP interventions but rather the link work programme (i.e. the referral mechanisms) so the
---

	authors should remove the term 'intervention' and say something like: 'the impact of the Link Worker programme on their health and wellbeing' It would also help – as raised previously – to have further details of the specific types of SP activities (referrals) undertaken and how frequently participants are engaging in these interventions; this is critical information as you might predict that participants who are not heavily engaged in activities/engage less frequently have a less positive experience of link workers, experience lower rates of behaviour change and potentially lower rates of perceived mental health. Levels of engagement with SP interventions (e.g. how many activities per week/month, duration of activities, etc) are likely to be strongly correlated with attitudes towards the impact of Link Workers so it is important to give some gauge as to the levels of involvement. If these data are unavailable this should be clarified in the methods section and identified as a shortcoming in the Discussion. The authors acknowledge that 'The sample was relatively small, but data saturation was reached' but I do not think it is possible to assert this given we don't know what the range of variation was in relation to frequency of engagement with SP interventions. I would add a note of caution along these lines to clarify that the study lacks key information about levels of participation in SP interventions.
--	---

REVIEWER	Richard Kimberlee University of the West England (Bristol), UK
REVIEW RETURNED	16-Feb-2017

GENERAL COMMENTS	I feel that the paper 's discussion is much improved and more in line with the ambitions posed by the authors. It still isn't clear what the referral criteria are for GPs. The reader can only assume that it is any long term condition. Thus things like housing advice or other social issues aren't part of the referral criteria. The reader still does not have a sense of how many patients have been seen by the Link Workers. Just the ambition of 11,000. It is good to see the profile of the sample but a comment on how representative this is of the patients using the service in its first 18 months of delivery would be useful. It might also be useful to use letters instead of numbers for the footnotes to the table so it doesn't confuse with the numbers for the references. Alternatively, the details could be included in a box attached to the table. However I feel that it still needs a thorough proof reading. Most of the references for this article are missing. The Social Prescribing Network should be capitalised. There are quite a lot of sentences that contain words that are joined together.
--

VERSION 2 – AUTHOR RESPONSE

Reviewer 1

1. The authors should remove the term 'intervention' and say something like 'the impact of the Link Worker programme on health and wellbeing'

As recommended by Reviewer 1, we now use the term Link Worker social prescribing (SP) programme throughout, which better reflects our investigation of the programme in its entirety. The model of social prescribing under investigation comprises two elements, (i) referral to a Link Worker who supports the individual to set and achieve goals that result in positive health changes and (ii) onward referral to other services. As such, we report data on the effects of the Link Worker SP

programme on health and health-related behaviours, as well as the effects of referral to specific services (additional data added see point 2 below). To clarify that we are evaluating the Link Worker social prescribing programme as a whole, we have revised the following: Abstract (p2) “To describe the experiences of patients with long-term conditions who are referred to and engage with a Link Worker social prescribing programme and identify the impact of the Link Worker programme on health and wellbeing”; study aims (p7) ‘to identify the impact of the Link Worker social prescribing programme on health and wellbeing’; and, discussion, (p22) ‘This study provides evidence about the impact of a Link Worker social prescribing programme on health and wellbeing’.

2. It would help to have details of the specific types of social prescribing (SP) activities (referrals) undertaken and how frequently participants are engaging in these interventions

On Table 1 (p12), we have added a column on the right hand side of the table which indicates the number of services that each participant was linked into. We have added Table 2 (p14) which shows the services by category and the total numbers of referrals into each service category. We do not have the data to indicate at individual level the amount of engagement with services which participants were referred into (e.g. number of activities per week/month and duration of activities). As recommended by Reviewer 1 we indicate this as a limitation of the study in the discussion section on Strengths, limitations and implications (p24-25), “the study lacks precise data about levels of participation in ‘linked’ services. We therefore cannot draw specific conclusions about the effects on health and wellbeing with respect to intensity and duration of ‘linked’ service use”.

3. Not possible to assert that ‘data saturation was reached’.

We have edited this section, and removed the reference to data saturation, but point out (p24) that there was a high degree of consistency in participant accounts of the impact of Link Worker social prescribing.

Reviewer 2

1. It isn't clear what the referral criteria are for GPs

The referral criteria are specified on p6, as follows, “Referral criteria are men and women aged 40-74 with one or more of the following long-term conditions: diabetes (type 1&2), chronic obstructive pulmonary disease, asthma, coronary heart disease, heart failure, epilepsy, osteoporosis, with or without anxiety or depression”.

2. The reader does not have a sense of how many patients have been seen by the Link Workers.

We state on p8, that, ‘The recruitment period was January to June 2016. Ways to Wellness was operational from April 2015, and in the 14 months since it started, 864 women and 739 men, average age 59, were referred from primary care and attended at least one Link Worker session’.

3. A comment on how representative the interview sample is of the patients using the service ... would be useful.

We do not claim that the 30 interview participants are ‘representative’ of all those using the service, as this is a qualitative study. Our strategy was to obtain a sample which varied as much as possible in order to identify a wide range of issues. On p8, we state the sampling criteria, “We set out to obtain a maximum variation sample across the four provider organisations based on the following criteria: age, gender; long-term condition; marital status, employment status, socio-economic status, and level of engagement with Ways to Wellness (intensive to non-intensive)”. In the discussion (p24-25) we also add as a limitation of the study, “... we cannot be certain of the extent to which the experiences of the 30 study participants reflect those of the 1,603 who used the service, although maximum variation sampling was used to include a wide range of participant experience”.

4. Use letters instead of numbers for footnotes to Table 1

We have done this

5. Needs thorough proof reading

The manuscript has been thoroughly proof read

6. Missing references

All references are now complete

7. Social Prescribing Network should be capitalised

This has been done (p6).

8. Sentences that contain words joined together

The quotes have been reformatted, and all are now presented separately.